# Mechanism of Antifungal Action of Monoterpene Isoespintanol against Clinical Isolates of *Candida tropicalis*

**DOI:** 10.3390/molecules27185808

**Published:** 2022-09-08

**Authors:** Orfa Inés Contreras Martínez, Alberto Angulo Ortíz, Gilmar Santafé Patiño

**Affiliations:** 1Biology Department, Faculty of Basic Sciences, University of Córdoba, Montería 230002, Colombia; 2Chemistry Department, Faculty of Basic Sciences, University of Córdoba, Montería 230002, Colombia

**Keywords:** isoespintanol, antifungal, *Oxandra xylopioides*, *Candida tropicalis*

## Abstract

The growing increase in infections by *Candida* spp., non-albicans, coupled with expressed drug resistance and high mortality, especially in immunocompromised patients, have made candidemia a great challenge. The efficacy of compounds of plant origin with antifungal potential has recently been reported as an alternative to be used. Our objective was to evaluate the mechanism of the antifungal action of isoespintanol (ISO) against clinical isolates of *Candida tropicalis*. Microdilution assays revealed fungal growth inhibition, showing minimum inhibitory concentration (MIC) values between 326.6 and 500 µg/mL. The eradication of mature biofilms by ISO was between 20.3 and 25.8% after 1 h of exposure, being in all cases higher than the effect caused by amphotericin B (AFB), with values between 7.2 and 12.4%. Flow cytometry showed changes in the permeability of the plasma membrane, causing loss of intracellular material and osmotic balance; transmission electron microscopy (TEM) confirmed the damage to the integrity of the plasma membrane. Furthermore, ISO induced the production of intracellular reactive oxygen species (iROS). This indicates that the antifungal action of ISO is associated with damage to membrane integrity and the induction of iROS production, causing cell death.

## 1. Introduction

Fungal infections constitute a continuing and serious threat to human health, especially in immunocompromised people, where the incidence of systemic candidiasis has increased substantially in recent years [1,2,3,4,5]. *Candida tropicalis* has emerged as one of the most important *Candida* spp., non-albicans, due to its high incidence in systemic candidiasis and its greater resistance to commonly used antifungals [6]. This yeast has been widely considered to be the second most virulent candida species, preceded only by *C. albicans* [7]; it is an opportunistic fungus that affects immunocompromised people and is capable of spreading to vital organs [8]. *Candida tropicalis* is recognized as a strong producer of biofilms, surpassing *C. albicans* in most studies; likewise, it produces a wide range of other virulence factors, including adhesion to epithelial cells, endothelial cells and other host surfaces as well as medical devices; secretion of lytic enzymes; and the so-called morphogenesis expressed by this yeast. *C. tropicalis* is a clinically relevant species and may be the second or third most important etiologic agent of candidemia, specifically in Latin American and Asian countries [7]. In Colombia, candidemia is a frequent cause of infection in the bloodstream, especially in Intensive Care Units (ICU); it represents 88% of fungal infections in hospitalized patients, with a mortality between 36% and 78%, and its incidence in Colombia is higher than that reported in developed countries and even in other Latin American countries [9]. It has been documented that *C. tropicalis* is associated with higher mortality compared to *C. albicans* and other non-albicans candida species, apparently showing a greater potential for dissemination in neutropenic individuals. This yeast is associated with malignancy, especially in patients who require prolonged catheterization, receive broad-spectrum antibiotics, or have cancer [10].

In this context, the search and development of new compounds with antifungal potential that are tolerable, effective and safe is urgent today. Various studies have reported the role of plants as a source of secondary metabolites with recognized medicinal properties [11], which can be used directly as bioactive compounds, as drug prototypes and/or as pharmacological tools for different targets [12]. Isoespintanol (ISO) (2-isopropyl-3,6-dimethoxy-5-methylphenol) is a monoterpene obtained for the first time from the aerial parts of *Eupatorium saltense* (Asteraceae) [13], and its synthesis was also reported [14]. Also, it has been extracted from *Oxandra xylopioides* (Annonaceae), whose antioxidant [15], anti-inflammatory [16], antispasmodic [17], vasodilator [18] and cryoprotectant in canine semen [19] effects have been reported, as well as its insecticide [20] and antifungal activity against *Colletotrichum* [21]. However, its antifungal potential against human pathogens has not been reported, so we hypothesize that ISO could have an effect against human pathogenic fungi such as *Candida* spp. The purpose of this research was to evaluate the antifungal activity of ISO, estimate its ability to eradicate mature biofilms and explore the mechanisms of action against clinical isolates of *C. tropicalis*, contributing to the search for new compounds of natural origin that can serve as adjuvants in the treatment of pathogenic yeasts resistant to antifungals.

## 2. Results

### 2.1. Obtaining and Identifying Isoespintanol

The ISO (1.2 g) was purified as a crystalline amorphous solid, with a purity greater than 99%, verified by GC-MS. Its structural identification by ^1^H-NMR, ^13^C-NMR, DEPT, COSY ^1^H-^1^H, HMQC and HMBC led unequivocally to propose the structure of 2,5-dimethoxy-3-hydroxy-*p*-cymene, isoespintanol, Figure 1. The EI-MS: [M] ^+^
*m*/*z* 210 (49%) and fragments *m*/*z* 195 (100%), 180, 165, 150, 135 and 91. ^1^H-NMR (CDCl_3_): δ 6.22 s, 1H (H6), δ 5.85 s, 1H (HO-3), δ 3.77 s, 3H (H12), δ 3.76 s, 3H (H11), δ 3.52 hep, J = 7.1 Hz, 1H (H8), δ 2.29 s, 3H (H7), δ 1.33 d, J = 7.1 Hz, 6H (H9-H10). ^13^C-NMR (CDCl_3_): δ 154.3 (C5), δ 147.4 (C3), δ 139.7 (C2), δ 126.8 (C1), δ 120.4 (C4), δ 104.4 (C6), δ 24.6 (C8), δ 60.8 (C11), δ 55.7 (C12), δ 20.6 (C9, C10), δ 15.8 (C7).

### 2.2. Phylogenomics of Candida tropicalis

Genome-wide based taxonomic study of the *C. tropicalis* clinical isolate confirmed the identification of this yeast. The results of the gDNA extraction, quantification and purity measurement (260/280) by spectrophotometry, as well as the general statistics of the NGS sequencing, the general statistics of the *C. tropicalis* genome assembly and the phylogenetic study, are shown in the Appendix A.

### 2.3. Antifungal Susceptibility Testing

ISO showed antifungal activity against all clinical isolates of *C. tropicalis* studied; we observed a reduction in the growth percentage of yeasts treated with ISO, compared to untreated isolates used as control. Figure 2 shows the similar tendency among the isolates to increase the percentage of growth reduction as the ISO concentration increases. All isolates showed FLZ MIC (MIC_90_) values ≥ 128 µg/mL. Table 1 shows the values of MIC (MIC_90_), MIC_50_ and MFC of the ISO; we observed ISO MIC values between 326.6 and 500 µg/mL and this effect on *C. tropicalis* was shown to be dependent on the ISO concentration.

We observed that after 48 h the values were between 350, 400, 450 and 500 µg/mL; as evidenced, the efficacy of ISO was different between strains of the same species. There was no visible growth of the yeasts in the presence of FLZ (MIC µg/mL) in SDA medium.

### 2.4. MTT Reduction Assay

The effect of ISO on the viability of *C. tropicalis* isolates was tested using the MTT reduction colorimetric method. As shown in Table 2, after treatment with different ISO concentrations, the viability of *C. tropicalis* cells had a significant reduction with increasing ISO concentration, compared to the control group (100% viability). We observed that at 125 µg/mL, the viability percentage was reduced by 56.3, 59.2, 57.4 and 67.0% for isolates 002, 003, 004 and 007, respectively, while at 250 µg/mL the viability percentage was reduced by 50, 48 and 51.7% for isolates 001, 005 and 006, respectively.

As shown in Figure 3, the viability of all isolates decreases by 50% at concentrations below 250 µg/mL.

### 2.5. Growth Inhibition Curve

The growth inhibition effect of ISO on *C. tropicalis* is shown in Figure 4. As observed in the two isolates of *C. tropicalis*, the number of yeasts began to proliferate rapidly at 6 h in the control group (cells untreated), unlike the yeasts treated with ISO where a significant delay in their growth was observed, beginning to proliferate after 12 and 24 h. After 24 h, the curve increased rapidly in these groups (this behavior was similar in all the isolates studied). The cells treated with FLZ (MIC µg/mL) showed in some cases a behavior similar to that of the cells treated with ISO, but in others the behavior was similar to the control group; no reduction in inhibition was observed with extension of time.

### 2.6. LIVE/DEAD Assay

Isolates of *C. tropicalis* treated with ISO and FLZ and without treatment were observed under fluorescence microscopy. AO diffuses across intact cytoplasmic membranes in living cells [22] where it interacts with DNA by emitting a bright green fluorescence [23]. In contrast, EB penetrates only cells with damaged membranes and cell walls in dead cells [24], intercalates with DNA, and emits a red-orange fluorescence. As seen in Figure 5, untreated cells with entirely green fluorescence grew well after 24 h (a), while dead cells with red fluorescence were massively observed in the group treated with ISO (b). The group treated with FLZ showed a lower proportion of dead cells (c).

### 2.7. Biofilm Reduction

All *C. tropicalis* isolates produced strong biofilms on polystyrene microplates as shown in Figure 6a. When the MIC of ISO was added to the biofilms formed from each isolate, a biofilm biomass eradication percentage between 20.3 and 25.8% was obtained after 1 h of exposure to ISO, while the percentage of biomass eradication of biofilms in cells treated with AFB was lower (7.2 and 12.4%).

### 2.8. Effect of ISO on Cell Membrane Integrity

Our results evidence the loss of membrane integrity, as shown in Figure 7, noticing a rare and insignificant PI fluorescence in untreated cells, but a considerable increase in PI fluorescence observed when the cells were treated with ISO in each isolate of *C. tropicalis* (Figure 7b). Evidencing differences in the intensity of fluorescence between the different isolates, the PI permeated percentage of cells treated with ISO for 12 h was between 1 and 39.1%, in all cases much higher than that of untreated cells (0.0–0.2); this indicates damage to the permeability of the membranes induced by ISO. In addition, this effect is different in each isolation. Cells treated with FLZ (Figure 7c) showed a significant decrease in fluorescence intensity compared to the fluorescence emitted by cells treated with ISO, with the percentage of fluorescence intensity of this group (0.1–0.5) being closest to that of the control (untreated cells).

### 2.9. Leakage of Nucleic Acids and Proteins through the Fungal Membrane

The action of ISO on the integrity of the membranes of *C. tropicalis* was also evaluated by release assays of intracellular constituents that absorb at 260/280 nm, such as nucleic acids and proteins. These assays were performed at 0, 30, 60 and 120 min after treatment with ISO MIC for each isolate. As seen in Figure 8, the OD_260_/OD_280_ values in the groups treated with ISO are significantly higher from time zero with a tendency to increase over time, compared to the groups treated with FLZ, in which a minimal and almost constant release of intracellular material was observed in all isolates of *C. tropicalis*. These results confirm fungal cell membrane damage caused by ISO.

### 2.10. Measurement of Extracellular pH

Extracellular pH measurements of *C. tropicalis* treated with ISO and FLZ and of untreated cells are shown in Figure 9. As observed in the four isolates shown, the cells treated with ISO showed an early significant increase in extracellular pH, compared with untreated cells (INO) in which the pH was observed to decrease. Cells treated with FLZ (MIC μg/mL) showed higher extracellular pH values compared to untreated cells, but in all cases these values were significantly lower compared to those of cells treated with ISO; these results confirm fungal cell membrane damage caused by ISO.

### 2.11. Transmission Electron Microscopy (TEM)

TEM was used to directly observe the effect of ISO on the morphology, intracellular changes and integrity of *C. tropicalis*. As shown in Figure 10a, in the untreated cells used as control, normal and intact cell morphology is observed, no damage to cell wall and membrane integrity is seen, there is no apparent disruption or release of intracellular content, and a homogeneous electronic density is observed at the level of the cytoplasm. However, after treatment with ISO (Figure 10c), the cells exhibited evident damage: irregular and deformed morphology, the cell membrane with holes and desquamation appearing partially dissolved, with loss of intracellular material and showing a heterogeneous electronic density with intense vacuolization and retraction at the level of the cytoplasm. In the cells treated with FLZ, some remained normal with intact cell walls and the damage at the membrane and cytoplasmic level was significantly less evident than that observed in cells treated with ISO.

### 2.12. Effect of Isoespintanol on the Production of Intracellular Reactive Oxygen Species (iROS)

Our results show that ISO treatment significantly increased the iROS load (represented by relative fluorescence intensity) in all *C. tropicalis* isolates compared to untreated cells (Figure 11) and to those treated with H_2_O_2_ (iROS-inducing control). The fluorescence intensity of DCFH-DA was different among the *C. tropicalis* isolates. Figure 12 shows differences in the fluorescence emitted by DCFH-DA from untreated, ISO-treated and H_2_O_2_-treated cells, evidencing an increase in iROS production in ISO-treated cells compared to the other two groups. The oxidative stress generated by the overproduction of iROS induced by ISO could damage intracellular components of these yeasts and induce cell death.

## 3. Discussion

Currently, the growing increase in candidemia caused by *Candida* spp., non-albicans, the resistance to conventional drugs used for its control, as well as the limitation of available antifungals, have made these infections a challenging problem in medical practice, stimulating the search for new molecules with antifungal properties. In this context, chemical compounds of natural origin become an excellent alternative.

In this research, we demonstrated that ISO, a monoterpene extracted from *O. xylopioides* leaves, has antifungal activity against clinical isolates of *C. tropicalis*, which were shown to be resistant to FLZ. The effects of ISO on various *C. tropicalis* cells were different, despite their being yeasts of the same species, and these results are consistent with those reported by [25] who found differences in the efficiency of the essential oil of *Ruta graveolens* against *Candida* spp., not only for different species but even among members of the same species. Previous studies have reported the antifungal activity of terpenes isolated from plants, indicating the influence of their chemical structure on their antimicrobial action; for example, the lipophilicity of monoterpenes has been shown to allow interaction with the fungal cell wall, facilitating their penetration of the cell membrane [26]. The antifungal activity of monoterpenes such as carvacrol [27], thymol [28,29], citral [30] and linalool [31], as well as compounds with chemical structures similar to ISO such as cinnamaldehyde [32] and eugenol, highlight its antifungal potential associated with damage to the integrity of the fungal plasma membrane and related to its lipophilic nature, a factor that increases fluidity and the permeability of the cell membrane of microorganisms; these compounds interfere with ion transport, causing an osmotic imbalance in the membrane and rendering its associated proteins ineffective, leading to inhibition of microbial growth and cell lysis [33]. This is consistent with our results. We report damage to the integrity of the fungal membrane of *C. tropicalis* caused by exposure to ISO, which is probably associated with the lipophilic nature of its structure. In addition, we do not rule out a possible interaction with ergosterol of the fungal membrane, taking into account the studies reported with thymol [34], a monoterpene very similar to ISO, indicating that the damage to the fungal membrane is associated with the interaction with yeast ergosterol. Results obtained by flow cytometry with PI (a fluorescent dye which can only enter cells that have permeable membranes, where it binds to nucleic acid and fluoresces red) [35], revealed changes in permeability of the fungal membrane caused by ISO. It is known that damage to the plasma membrane can cause the collapse of the electrochemical potential, due to the formation of pores [28], where the loss of osmotic balance and the entry of ions and liquids, as well as the loss of the cytoplasmic content of cells such as soluble proteins, carbohydrates and ribonucleic acids, make the cell unable to self-regulate, resulting in cell death [36]. This is consistent with our results, which show a significant and early release of intracellular material (at 260/280 nm) in yeasts treated with ISO, with a tendency to increase over time compared to untreated cells; in addition, the evident loss of intracellular protons (demonstrated by increased extracellular pH) confirmed the compromise of permeability in the plasmatic membrane of these ISO-treated pathogens.

In addition, the damage to the membrane integrity of *C. tropicalis* by ISO was also evidenced by TEM, which showed damage to the morphology and envelope of yeasts treated with ISO; these relevant morphological abnormalities are of great importance since they can impede the growth, viability and virulence of these yeasts, as suggested by studies with eugenol and *Trichophyton rubrum* [33]. The morphological changes could be associated with the inhibitory effect shown by ISO on mature biofilms of *C. tropicalis*, taking into account what was reported by [37], who indicates the inhibitory effect on the development of *C. tropicalis* biofilms due to morphological changes caused by the drug. On the other hand, fluorescence microscopy with AO/EB confirmed the damage to the cell membrane, which has been reported as a target point for the antimicrobial action of terpenes, which, due to their lipophilic nature, can interact with it causing its expansion, increased fluidity and permeability, with consequent damage to its structure and function [38], which is consistent with the results reported in this study.

On the other hand, the results found show that ISO stimulates the production of iROS in *C. tropicalis*; similar results were reported in previous studies with thymol [29]. The regulated synthesis of ROS by specific fungal NADPH oxidases plays a key role in fungal cell differentiation and development. In low concentration, they are an important intracellular messenger in many molecular events and play a key role in host defense [39], but in large quantities, it is well known that oxygen radicals can rapidly lead to the disintegration of biological membranes, resulting in cell death. iROS accumulation causes oxidative damage to mitochondrial proteins that appear to be disproportionately affected under oxidative stress, inducing mitochondrial membrane potential collapse [36], which in turn leads to increased iROS generation [40]. Likewise, it has been reported that the accumulation of iROS is necessary and sufficient to induce apoptosis in yeast, its presence being one of the first changes involved in this type of cell death [41]. High levels of iROS can cause oxidative stress in yeast due to the formation of oxidized cellular macromolecules, including lipids, proteins and nucleic acids, thus triggering the onset of apoptosis [42] and loss of viability [43]. Our results show that treatment with ISO significantly increased the load of cells with high levels of iROS production in all clinical isolates of *C. tropicalis*, showing different fluorescence intensities of DCFH-DA depending on the strain studied, compared to untreated strains where no fluorescence was observed; in addition, the fluorescence emitted in the cells treated with ISO was much higher in contrast to the fluorescence emitted in the cells treated with H_2_O_2_ (ROS-inducing control) in all cases. Some studies have reported the antioxidant capacity of ISO in oily matrices, as well as its free radicals scavenging capacity [44] and its participation in the oxidative stabilization of palm olein [45]; however, our studies with pathogenic yeasts differ from these results, showing the ability of ISO to induce the production of iROS in isolates of *C. tropicalis*. This leads us to think that the oxidative stress generated by the overproduction of iROS induced by ISO could damage intracellular components of these yeasts and induce cell death, and this could be another mechanism of antifungal action of ISO against *C. tropicalis*.

*Candida tropicalis* is well known for its ability to form strong biofilms, which vary depending on the origin of the infection [46]; these biofilms associate these pathogens with high mortality, possibly due to the low permeability of the matrix to conventional antifungal drugs [47]. The *C. tropicalis* isolates in this study were strong biofilm producers, consistent with studies reported by [37], who reported strains of *C. tropicalis* with strong, fast-growing biofilms as a result of their high metabolic activity. Comparing the efficacy of ISO with AFB, we highlight the role of ISO in the eradication of mature biofilms during 1 h of treatment (between 20.3 and 25.8%), and in all cases it was greater than with AFB (between 7.2 and 12.4%); this is consistent with the studies reported by [37] indicating the ability of liposomal AFB to inhibit further biofilm growth, but its ineffectiveness in eradicating mature biofilms, even at high doses. It is important to highlight that natural compounds obtained from plants can potentially be used to combat multiresistant and biofilm-forming strains of *Candida* spp., thus becoming a promising alternative to antifungal drugs [48]. In addition, the cytotoxicity of ISO on human peripheral blood lymphocytes [49] and murine macrophages (RAW 264.7) [45] has been reported, indicating that this compound does not have genotoxic or cytotoxic effects on these cells at the concentrations tested. Our results add new and important information about the antifungal potential of ISO monoterpene, showing more than one target of action on *C. tropicalis* cells; in addition, we provide information that serves as a basis for future research in the exploration of other possible targets of antifungal action of this monoterpene that could serve as adjuvants for the therapy of infections by these pathogenic yeasts.

## 4. Materials and Methods

### 4.1. Reagents

RPMI 1640, phosphate buffered saline (PBS), and yeast peptone dextrose broth (YPD) were obtained from (Thermo Fisher Scientific, Waltham, MA, USA); 3-N-morpholinopropanesulfonic acid (MOPS) was obtained from (Merck); propidium iodide (PI), 2′,7′-dichlorofluorescin diacetate (DCFH-DA), potato dextrose broth (PDB), sabouraud dextrose agar (SDA), sabouraud dextrose broth (SDB), amphotericin B (AFB), acridine orange (OA), ethidium bromide (EB), and crystal violet used in this study were obtained from Sigma-Aldrich, United States; glacial acetic acid was obtained from Carlo Erba Reagents, Italy; and Fluconazole (FLZ) was obtained from Pfizer.

### 4.2. Obtaining and Identification of Isoespintanol

ISO was isolated from leaves of *O. xylopioides*, collected in October 2019 from a specimen located in the Municipality of Monteria, Department of Córdoba, with coordinates 08°48′17″ north latitude and 75°42′07″ west longitude. An herbarium specimen is deposited in the Joaquin Antonio Uribe Botanical Garden of Medellin, Colombia, with the collection number JAUM 037849. The ISO was obtained by hydrodistillation and crystallization, from 5 g of petroleum benzyne extract of the leaves of *O. xylopioides*, following the methodology described in a previous work [50], with some modifications that included successive crystallizations with n-hexane that led to obtaining 1.2 g of the pure compound. The purity was verified using a gas chromatograph coupled to a Thermo Scientific model Trace 1310 mass spectrometer, with an AB-5MS column, (30 m × 0.25 mm i.d. × 0.25 μm). The temperature gradient system started at 80 °C for 10 min up to 200 °C at 10 °C/min. The temperature was increased to 240 °C at 4 °C/min and finally it was brought up to 290 °C for 10 min at 10 °C/min. The injection was splitless type, with an injection volume of 1 μL. The mass spectrum was obtained in electron impact ionization mode at 70 eV. The structure of the ISO was determined using ^1^H-NMR, ^13^C-NMR, DEPT, COSY ^1^H-^1^H, HMQC and HMBC spectra, performed on a 400 MHz Bruker Advance DRX spectrometer, in deuterated chloroform (CDCl_3_).

### 4.3. Strains

Seven clinical isolates of *C. tropicalis* (001 to 007) were used in this study. The isolates were cultured from blood culture and tracheal aspirate samples from hospitalized patients at the Salud Social S.A.S. from the city of Sincelejo, Colombia. All microorganisms were identified by standard methods: Vitek 2 Compact, Biomerieux SA, YST Vitek 2 Card and AST-YS08 Vitek 2 Card (Ref 420739). SDA medium and BBL CHROMagar Candida medium were used to maintain the cultures until the tests were carried out. The identification of one of the *C. tropicalis* isolates was confirmed through a genome-wide taxonomic study.

### 4.4. Phylogenomics of Candida tropicalis

#### 4.4.1. Extraction of Genomic DNA from *Candida tropicalis*

Colonies of *C. tropicalis* in SDA were used for genomic DNA (gDNA) extraction using the Qiagen DNeasy PowerLyzer PowerSoil kit, following the manufacturer’s instructions. The extracted gDNA was quantified by light absorption at 260 nm using the NanoDrop™ 2000-Thermo Scientific™ and frozen at −20 °C for subsequent genomic sequencing experiments.

#### 4.4.2. WGS (Whole Genome Shotgun) Genomic Sequencing of *Candida tropicalis* on the Illumina Novaseq Platform

The sequencing of the gDNA extracted from *C. tropicalis* was carried out using Truseq Nano DNA libraries (350) and the Illumina NovaSeq platform, through which 150-base paired reads were generated. Subsequently, the assembly of the *C. tropicalis* genome was carried out using the SPADES program. For phylogenetic analyses, conserved single-copy genes were used. Afterwards, these genes were aligned and concatenated with MAFFT; the iqTREE software was used to select the substitution models and generate the tree (Maximum Likelihood), which was visualized with FIGTREE.

### 4.5. Antifungal Susceptibility Testing

The minimum inhibitory concentration (MIC) of ISO against clinical isolates of *C. tropicalis* was defined as the lowest concentration at which 90% (MIC_90_) of fungal growth was inhibited, compared to the control. MIC was established by performing broth microdilution assays, using 96-well microtiter plates (Nunclon Delta, Thermo Fisher Scientific, Waltham, MA, USA), as described in *Clinical Laboratory Standards Institute* (CLSI) method (M27-A3) [51] and *The European Committee for Antimicrobial Susceptibility Testing* (EUCAST) [52], with minor modifications. Serial dilutions were made in RPMI 1640 broth (pH 7.0) buffered with 0.165 M MOPS, to obtain final concentrations of 31.25 to 1000 µg/mL of ISO in each reaction well. Stock solutions of ISO at 20,000 µg/mL in DMSO and FLZ at 1500 µg/mL in 10% DMSO in distilled water were prepared. Assays were performed with a final volume of 200 µL per well as follows: 100 µL of fungal inoculum at a concentration of 10^6^ CFU/mL and 100 µL of ISO adjusted to achieve the concentrations described above in a final reaction system. Isolates of *C. tropicalis* without ISO and with FLZ were used as growth controls and positive controls, respectively; wells with culture medium without inoculum and without ISO were used as negative controls. For each test, controls were made with the different concentrations of ISO in culture medium without inoculum. The plates were incubated at 37 °C for 24 h. Inhibition of fungal growth by ISO was determined by change in optical density using a SYNERGY LX microplate reader (Biotek), at 530 nm, from the start of incubation to the end time (24 h), and the reduction percentage of growth was calculated [53] using the following equation:%Reduction = (1 − (OD_t24_ − OD_t0_/OD_gc24_ − OD_gc0_)) × 100
where, OD_t24_: optical density of the test well at 24 h post-inoculation; OD_t0_: optical density of the test well at 0 h post-inoculation; OD_gc24_: optical density of the growth control well at 24 h post-inoculation; OD_gc0_: optical density of the growth control well at 0 h post-inoculation.

Subsequently, the minimum fungicidal concentration (MFC) was determined by taking 10 µL from each well and inoculating it onto SDA. The plates were sealed and incubated at 37 °C for 24/48 h. MFC was considered the lowest concentration capable of inhibiting 99% of yeasts [25]. All experiments were performed in triplicate.

### 4.6. MTT-Reduction Assay

To evaluate the cell viability of *C. tropicalis* in the presence of ISO, the colorimetric assay for the reduction of MTT (3-(4,5-dimethylthiazol-2-yl)-2,5-diphenyl-2H-tetrazolium) was performed as described by [54]. MTT was dissolved in PBS at 2.5 mg/mL and filtered. The fungal inoculum at a cell concentration of 10^6^ CFU/mL was inoculated into 96-well plates at the previously described ISO concentrations, and the plates were incubated for 24 h at 37 °C. Cells with medium were used as control. Then 50 µL of the MTT solution was added to the cells under evaluation at a concentration of 500 µg/mL. The plates were incubated in the dark for 4 h at 37 °C. The tests were performed in triplicate. Viable cells with metabolic activity convert MTT (yellow color) to formazan, which was solubilized with dimethyl sulfoxide (DMSO) showing a purple color; for this, 50 µL of supernatant is removed from each well and 50 µL of DMSO is added, which was read at a wavelength of 550 nm in a SYNERGY LX microplate reader (Biotek) after shaking in the equipment.

### 4.7. Growth Inhibition Curves

The growth inhibition curve of *C. tropicalis* by ISO was performed following the methodology proposed by [23], with minor modifications. Isolates of *C. tropicalis* were cultured in SDB medium for 24 h at 37 °C. The fungal inoculum was standardized until reaching a cell concentration of 10^6^ CFU/mL in glass tubes, then the ISO was added (the MIC for each isolate) and the tubes were incubated at 37 °C for 48 h. Subsequently, 1 mL was taken from each tube at times 0, 2, 4, 8, 12, 24, 36 and 48 h and read at 530 nm in a SYNERGY LX microplate reader (Biotek). Tubes with the fungal inoculum and FLZ were used as controls. All assays were performed in triplicate.

### 4.8. LIVE/DEAD Assays

The LIVE/DEAD assays were developed following the methodology proposed by [23], with some modifications. A suspension of *C. tropicalis* (10^6^ CFU/mL) was placed on sterile slides and incubated for 24 h. The cells were then washed three times with PBS. Subsequently, the ISO MIC for each yeast and FLZ were added to the experimental groups and the fungal inoculum in RPMI 1640 broth was used as a control. Prepared slides were incubated at 37 °C for 24 h and then washed three times with PBS. Together, AO (5 µL, 100 mg/L) and EB (5 µL, 100 mg/L) were mixed under dark conditions and added to slides under dark conditions for 30 s. Next, the samples were observed in an Olympus BX43 fluorescence microscope and photographed with a DP72 camera.

### 4.9. Quantitative Assessment of Biofilm Formation

Isolates of *C. tropicalis* were evaluated to quantify the reduction of biofilms in the presence of ISO following the methodology reported by [25], with some modifications. For biofilm formation, yeast colonies in SDA with 24 h of incubation were used to standardize the inoculum until reaching a concentration of 10^6^ cells/mL. Then, in 96-well plates, 200 µL of the fungal inoculum was cultured in each well in YPD broth and incubated at 37 °C for 48 h. Then the broth was removed from the microplates and 200 µL of the ISO MIC for each isolation in YPD broth was added and incubated at 37 °C for 1 h. Then, the floating cells were removed and the biofilm at the bottom of the wells was washed with deionized water three times. Six replicates of each sample were made. Cultures without ISO were taken as control and AFB was used as positive control. Biofilm reductions were quantified by staining wells with 0.1% crystal violet for 20 min. The samples were washed with deionized water until the excess dye was removed. Finally, the samples were soaked in 250 µL of 30% glacial acetic acid. Absorbance values were measured at 590 nm (OD_590_) using a SYNERGY LX microplate reader (Biotek). Biofilm production was grouped into the following categories: OD_590_ < 0.1: non-producers (NP), OD_590_ 0.1–1.0: weak producers (WP), OD_590_ 1.1–3.0: moderate producers (MP) and OD_590_ > 3.0: strong producers (SP). Biofilm reduction was calculated using the following equation:% Biofilm reduction: AbsCO − AbsISO/AbsCO × 100
where, AbsCO: absorbance of the control and AbsISO: absorbance of the sample treated with ISO.

### 4.10. Effect of Isoespintanol on Cell Membrane Integrity

To evaluate the effect of ISO on cell membrane integrity, the methodology proposed by [35] was used, with minor modifications. The fungal cells (10^6^ CFU/mL) were suspended in RPMI 1640 medium and treated with ISO MIC for each isolation and incubated for 12 h at 30 °C. Cells without ISO and cells treated with FLZ (100 µg/mL) were used as controls. Subsequently, the cells were incubated with 1.49 µM PI in water at 30 °C for 50 min. Then, the cells were collected by centrifugation (3000× *g* 10 min, 4 °C), resuspended in PBS and finally analyzed by flow cytometry (20,000 events analyzed per assay), using the BD FACS CANTO II flow cytometer and analyzed with the BD FACS DIVA software. The excitation and emission for PI were 488 nm and 630 nm, respectively. All experiments were performed in triplicate.

### 4.11. Leakage of Nucleic Acids and Proteins through the Fungal Membrane

The release of intracellular material was measured according to the methodology proposed by [30], with some modifications. Yeasts grown in SDB were centrifuged at 3000× *g* for 20 min, washed three times and resuspended in 20 mL of PBS (pH 7.0). Then, the cell suspension was treated with ISO (MIC for each isolation) and incubated at 37 °C for 0, 30, 60 and 120 min. Subsequently, 2 mL of the samples were collected and centrifuged at 3000× *g* for 20 min. Then, to determine the concentration of the released constituents, 2 mL of supernatant was used to measure the absorbance at 260/280 nm with the Spectroquant^®^ Prove 300 UV/Vis spectrophotometer. Samples without ISO and samples with FLZ were used as controls. All assays were performed in triplicate.

### 4.12. Measurement of Extracellular pH

The measurement of extracellular pH of *C. tropicalis* after treatment with ISO was determined according to [30], with some modifications. 100 µL of the yeast suspension (10^5^ CFU/mL) was added to 20 mL of SDB and incubated at 37 °C for 48 h. Then, the samples were centrifuged at 3000× *g* for 20 min; the pellet was collected, resuspended, and washed three times with bidistilled water and resuspended again in 20 mL of sterile bidistilled water. After the addition of ISO (MIC of each isolate), the extracellular pH of *C. tropicalis* was determined at 0, 30, 60 and 120 min, using a Schott^®^ Instruments Handylab pH 11 pHmeter. Samples without ISO and samples with FLZ were used as controls.

### 4.13. Transmission Electron Microscopy (TEM)

The morphology of *C. tropicalis* after ISO treatment was analyzed by TEM. The concentration of *C. tropicalis* was adjusted to 10^6^ CFU/mL; the suspension was mixed with ISO (200 µg/mL) and incubated at 37 °C for 24 h. Subsequently, the cells were collected and fixed in 2.5% glutaraldehyde in phosphate buffer pH 7.2 at 4 °C; they were centrifuged at 13,000 rpm for 3 min and the button at the bottom of the vial was postfixed in 1% osmium tetroxide in water for 2 h at 4 °C. Then, pre-imbibition with 3% uranyl acetate was performed for 1 h at room temperature, after which the cells were dehydrated in an ethanol gradient (50% for 10 min, 70% for 10 min, 90% for 10 min, 100% for 10 min), acetone-ethanol (1:1) for 15 min and embedded in SPURR epoxy resin. The samples were cut in a Leica EM UC7 ultramicrotome, at 130 nm thickness, and contrasted with 6% uranyl acetate and lead citrate, and then finally observed in a JEOL 1400 plus transmission electron microscope. The photographs were obtained with a Gatan Orius CCD camera.

### 4.14. Effect of Isoespintanol on the Production of Intracellular Reactive Oxygen Species (ROS)

The detection of intracellular ROS (iROS) was carried out according to the protocol described by [55], with minor modifications. Fungal cells (10^6^ CFU/mL) were incubated in PDB with the MIC of the ISO for each isolate for 24 h at 35 °C. A cell suspension under the same conditions without ISO was used as a negative control. The cells were then incubated with 20 µM of DCFH-DA for 30 min in the dark at 35 °C. Afterwards, the cells were collected, washed, resuspended in PBS and analyzed by flow cytometry. The excitation and emission for DCFH-DA were at 485 and 535 nm. H_2_O_2_ was used as iROS-inducing positive control [56].

### 4.15. Data Analysis

The results were analyzed using the statistical software R version 4.1.1 and the Excel program. Initially, the Shapiro Wilk test was used to find out the distribution of the data. Subsequently, the Pearson correlation coefficient was used to measure the degree of linear correlation between ISO concentration and percentage reduction of fungal growth. To compare the effects of ISO and AFB on the reduction of biofilms, the Tukey test was used; this test was also used to compare the effects of ISO and FLZ on the leakage of intracellular material through the membrane (260/280 nm). The Kruskal–Wallis test was used to compare the effects of the treatments on the extracellular pH of *C. tropicalis*.

## 5. Conclusions

In this study, we investigated the antifungal effect of ISO against clinical isolates of *C. tropicalis*, as well as its role in biofilm disruption. In addition, we explored the mechanisms of action presented by this monoterpene. Our study shows antifungal action of ISO against these pathogenic yeasts, this effect being associated with damage to the plasma membrane and the induction of iROS production, in addition to its action against fungal biofilms, showing that ISO has more than one cellular target in its antifungal potential.

## Figures and Tables

**Figure 1 molecules-27-05808-f001:**
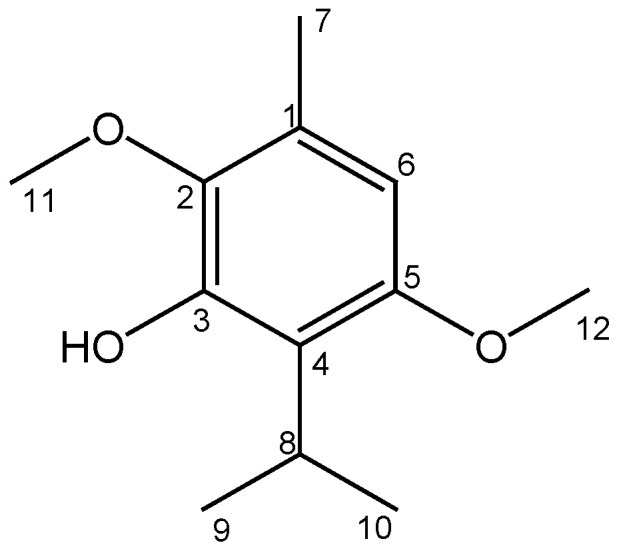
Structure of Isoespintanol.

**Figure 2 molecules-27-05808-f002:**
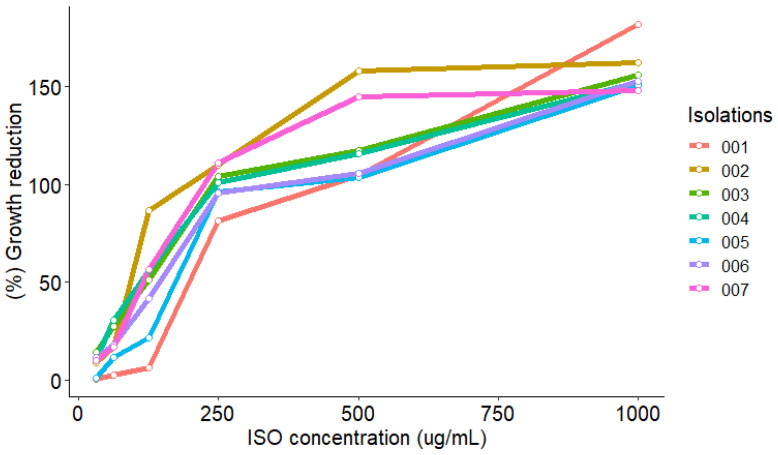
Growth reduction of *C. tropicalis* isolates exposed to ISO (MIC of each isolate). A strong and positive linear relationship between the ISO concentration and the yeast growth reduction percentage is observed, i.e., as the ISO concentration increases, the yeast growth reduction percentage also increases, which coincides with the Pearson correlation coefficient (0.83 < r < 0.96) in all isolates. In addition, the hypothesis test on the correlation coefficient yields a *p*-value < 0.05, which indicates that, with 95% confidence, there is a significant linear relationship.

**Figure 3 molecules-27-05808-f003:**
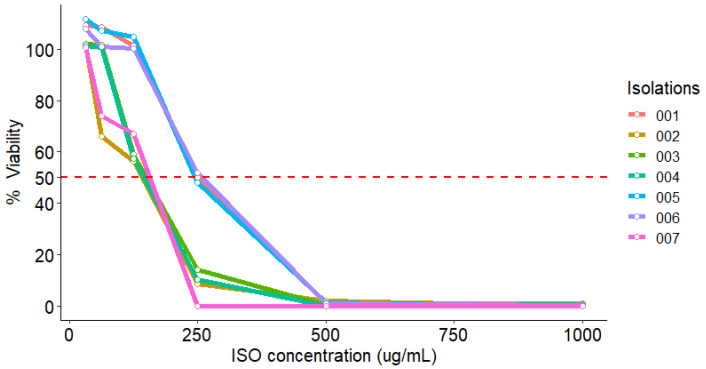
MIC_50_ of viability percentage of *C. tropicalis* isolates exposed to ISO (MIC for each isolate).

**Figure 4 molecules-27-05808-f004:**
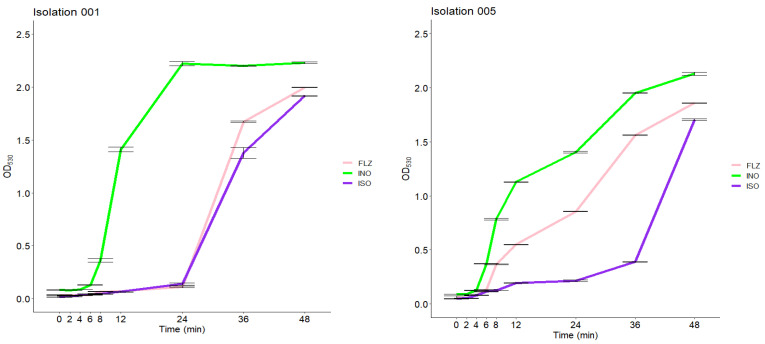
Inhibitory effect of ISO on *C. tropicalis* (001 and 005), at different times. The delay in the growth of *C. tropicalis* treated with ISO is observed, unlike the untreated cells (INO). The behavior of the cells treated with FLZ (MIC µg/mL) is also observed.

**Figure 5 molecules-27-05808-f005:**
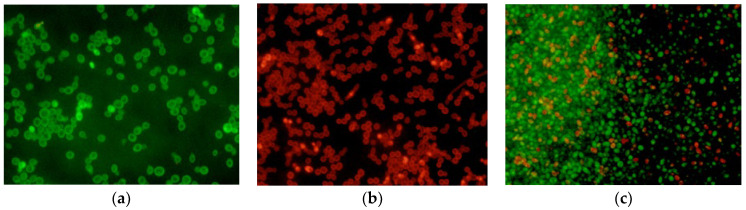
Fluorescence microscopy of *C. tropicalis* without treatment (**a**), treated with ISO (**b**) and treated with FLZ (**c**) after 24 h. Live cells with intact membranes appear green, while dead cells with damaged membranes appear red.

**Figure 6 molecules-27-05808-f006:**
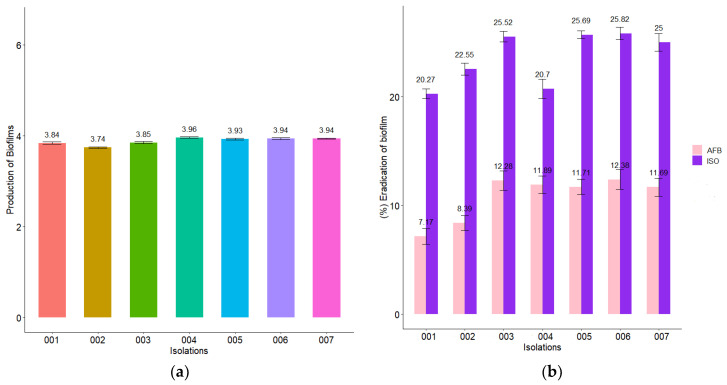
Action of ISO and AFB on *C. tropicalis* biofilms. (**a**) Biofilm formation at 37 °C for 48 h, where OD_590_ > 3 indicates strong biomass production in biofilms. (**b**) Percentage reduction of biofilms after 1 h of treatment with ISO MIC for each isolate and AFB (50 µg/mL). The results of the ANOVA showing a value of *p* < 0.05 and the Tukey test with a confidence level of 95% indicate that there is a significant difference between the effect of ISO and the effect of AFB on the reduction of biofilms.

**Figure 7 molecules-27-05808-f007:**
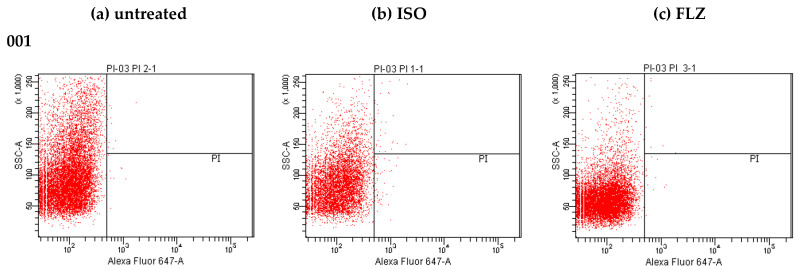
Changes in membrane permeability of clinical isolates of *C. tropicalis* treated with ISO: PI staining for detection of membrane permeability disruption in *C. tropicalis*. (**a**) untreated cells, (**b**) cells treated with ISO, and (**c**) cells treated with FLZ (MIC µg/mL).

**Figure 8 molecules-27-05808-f008:**
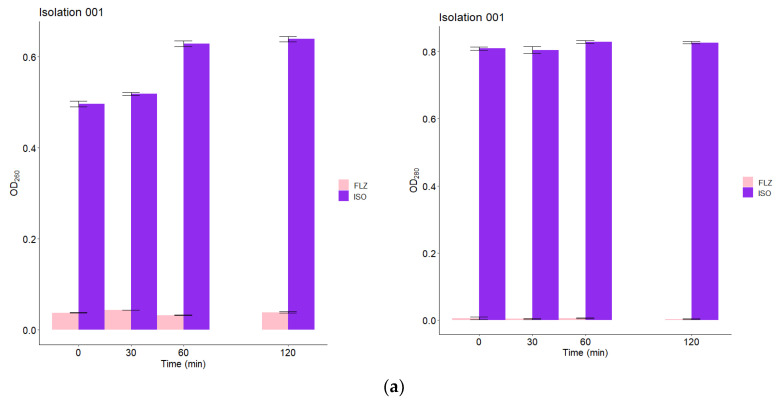
Release of intracellular content at 260/280 nm versus time of *C. tropicalis* treated with ISO (MIC µg/mL) and FLZ (MIC µg/mL). The figure shows the OD260/OD280 values of isolate 001 (**a**) and isolate 002 (**b**) treated with ISO and FLZ at different times. Results are expressed as the absorbance of the sample (treated with ISO) minus the absorbance of the control (samples without ISO). The results of the ANOVA and the Tukey test with a confidence level of 95% show statistically significant differences between the effect of ISO and the effect of FLZ on the leakage of intracellular material from all *C. tropicalis* isolates in this study.

**Figure 9 molecules-27-05808-f009:**
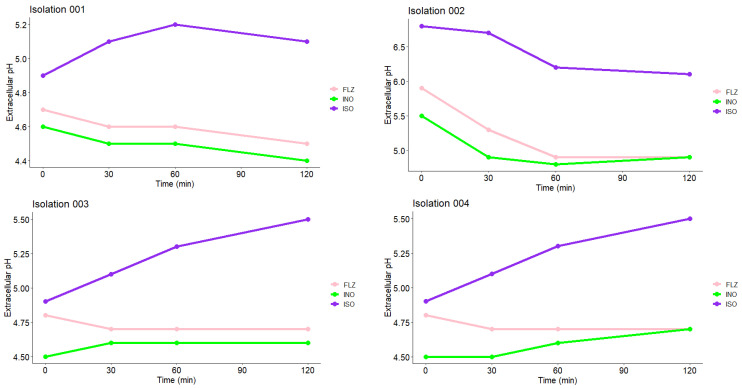
Measurement of extracellular pH of *C. tropicalis* treated with ISO, FLZ and untreated cells (INO). According to the results obtained by the Kruskal-Wallis test with a value of *p* = 0.05, there are significant differences between the treatments on the extracellular pH output of the yeasts.

**Figure 10 molecules-27-05808-f010:**
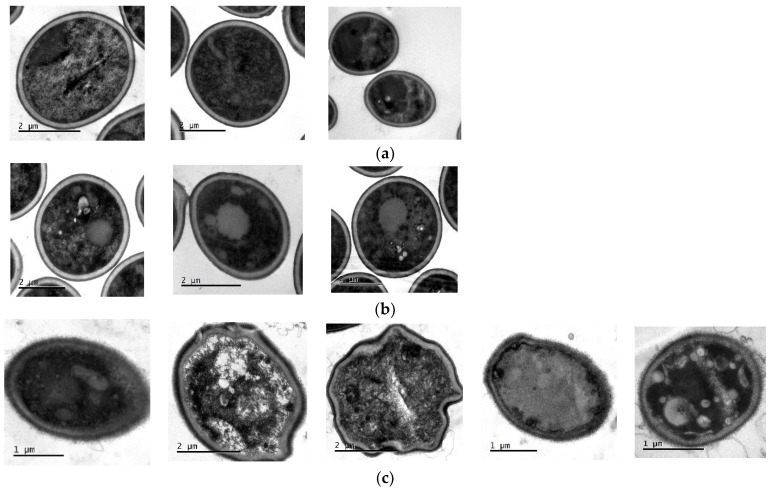
TEM of *C. tropicalis* untreated (**a**), treated with FLZ (**b**) and treated with ISO (**c**). The evident change in the morphology of the cells treated with ISO, as well as the damage to the integrity of the fungal cell membrane, is observed.

**Figure 11 molecules-27-05808-f011:**
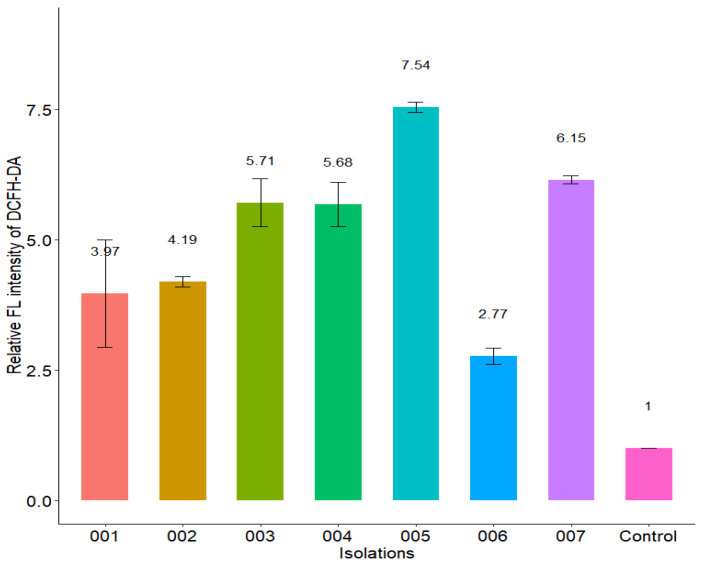
Effect of ISO treatment on iROS production in *C. tropicalis*. The relative fluorescence intensity of the isolates treated with the ISO MIC for each one was compared with that of the control group (untreated cells).

**Figure 12 molecules-27-05808-f012:**
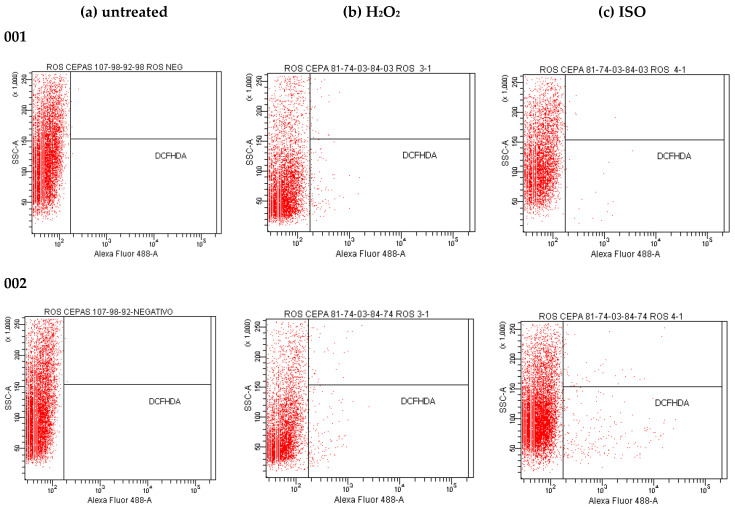
DCFH-DA fluorescence emitted by *C. tropicalis* yeasts untreated (**a**), treated with H_2_O_2_ (**b**) and treated with ISO (**c**).

**Table 1 molecules-27-05808-t001:** Minimum inhibitory concentration (MIC) and minimum fungicidal concentration (MFC) values (µg/mL) of ISO and FLZ against *C. tropicalis*.

*C. tropicalis*	ISO			FLZ
	MIC_90_	MIC_50_	MFC	MIC_90_
CLI 001	470	261.2	500	875.1
CLI 002	326.6	59.38	350	751
CLI 003	413.3	124.4	400	875.1
CLI 004	420.8	121.5	450	751
CLI 005	500	234.6	500	256
CLI 006	463.9	179.8	450	128
CLI 007	391.6	107	400	751

**Table 2 molecules-27-05808-t002:** Viability percentages with MTT.

Isolations							
ISO (µg/mL)	001	002	003	004	005	006	007
**31.25**	109.1	101.8	102.0	101.3	111.7	107.7	100.6
**62.5**	108.6	65.9	101.7	100.6	107.1	100.8	74.0
**125**	101.3	56.3	59.2	57.4	104.6	100.1	67.0
**250**	50.0	8.7	14.2	10.2	48.0	51.7	0.0
**500**	0.9	1.8	0.3	0.0	1.3	0.9	0.0
**1000**	1.1	0.0	0.8	0.6	0.0	0.2	0.0
**MIC_50_**	236.8	103.1	141.8	136.4	230.9	235.2	113.7

## Data Availability

The data presented in this study are available in the article and the Appendix A.

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
