# Peer review of "Mechanism of Antifungal Action of Monoterpene Isoespintanol against Clinical Isolates of Candida tropicalis"

_molecules, 2022, doi:10.3390/molecules27185808_

Round 1

Reviewer 1 Report

In the present manuscript “Mechanism of antifungal action of monoterpene isoespintanol against clinical isolates of Candida tropicalis” is evaluated the antifungal activity of the monoterpene isoespintanol (ISO), isolated from leaves of Oxandra xylopioides, against seven different clinical isolates of C. tropicalis by performing broth microdilution as described in method M27-A3 of the Clinical Laboratory Standards Institute (CLSI). In addition, the antifungal action of the compound on preformed biofilm was evaluated. Finally, the mechanism of action of ISO was studied, evaluating the (1) effect of ISO on cell membrane integrity (PI staining and flow cytometry analysis); (2) the release of intracellular material; (3) extracellular pH; (4) morphological changes of yeast using TEM and (5) production of intracellular ROS.  The work is very complete, with a good experimental design and presents novel results. It also presents adequate references mostly recent publications. Therefore, I think it deserves to be published in this journal after making the following corrections:

1.    Introduction

-  In line 3, after “in recent years”, change “,” to “.” to make the sentence shorter.

-  The first time the compound isoespintanol is mentioned it must go with the full name, and then if you use an abbreviation (ISO).

-  Add the plant family of Eupatorium saltense.

2. Materials and methods

- Add month and year of collection of the plant Oxandra xylopioides.

- In the section of strains used for the tests, mention the numbers of strains used (001 to 007).

- In the section “Antifungal Susceptibility Testing”:

(1) Change “IC90” to “MIC90”

(2) In reference number 23 cite the one corresponding to the document M27-A3. The correct reference is: Clinical and Laboratory Standards Institute. 2008. Reference method for broth dilution antifungal susceptibility testing of yeasts; approved standard, 3rd ed. CLSI document M27-A3. Clinical and Laboratory Standards Institute, Wayne, PA.

(3) Add in the manuscript the formula used to calculate the percentage of growth reduction.

(4) Is section “2.6 Determination of Minimum Fungicidal Concentration (MFC)” correct? because in the previous section the determination of fungicidal activity is described, the information is repeated.

(5) Was a negative control performed (culture medium without fungus and without compound)? controls of the different concentrations of the compounds in culture medium without inoculum were carried out? Add this information.

(6) Were stock solution of ISO and FCZ prepared (from which the solutions to be tested are prepared)? at what concentrations? Was DMSO used?

- In the section “MTT-Reduction Assay”:

(1) Is it done to analyze cell viability in the same microplates where the ISO MIC was tested?

(2) Determinations are made in duplicate, triplicate??? add number of replicas.

(3) Why is MTT added to a concentration of 500 ug/ml, If 50 ul of a 5 mg/ml solution is taken in a final volume of 250 ul.

(4) Once the formazan precipitate is formed, do you remove the culture medium and then add DMSO? better explain in this part the staining procedure.

- In the section “Growth Inhibition Curves”:

(1) Why is the fungus incubated at 35°C and not at 37°C?

(2) Determinations are made in duplicate, triplicate??? add number of replicas.

- In the section “Quantitative Assessment of Biofilm Formation”:

(1) Change “200 ul of the samples” to “200 ul of the fungal inoculum”.

(2) What concentration of inoculum was used for this assay? How was it prepared? Please, add this information.

(3) After incubation of the cells for 48 h, the culture medium is removed, washes were performed with PBS to eliminate the planktonic cells?

(4) After the formula of % biofilm reduction, change “control sample” to “control” (which corresponds to the control of biofilm formation without treatment).

- In the section “Leakage of Nucleic Acids and Proteins Through the Fungal Membrane”:

(1) Determinations are made in duplicate, triplicate??? add number of replicas.

3. Results

- In the section “Obtaining and Identifying Isoespintanol”:

(1) Add the chromatogram and the mass spectrum obtained by GC-mass to confirm purity (as supplementary material).

- In the section “Phylogenomics of Candida tropicalis”:

(1) Were all seven isolates evaluated or only one?

(2) In figure S3 I do not find the evaluated strains of C. tropicalis.

- In the section “Antifungal Susceptibility Testing”:

(1) Change “IC50 or IC90” to “MIC50 or MIC90”.

(2) In figure 2: Why are there percentage values of growth reduction greater than 100? Add the error bar (standard deviation) in the determinations.

 (3) In table 1: Change “,” to “.” in the values.

- In table 2: Change “,” to “.” in the values, and change “IC50” to “MIC50”.

- In figure 3: Add the error bar (standard deviation) in the determinations, and change “IC50” to “MIC50”.

- In figure 4: Add the error bar (standard deviation) in the determinations, and in materials and methods it says that the determinations are made at 600nm and in the graph it appears at 530nm, what is the correct OD?

- In figures 6, 8 and 11: Add the error bar (standard deviation) in the determinations.

- In the section “Transmission Electron Microscopy (TEM)”:

(1) Change “(Figure 10 (b))” to “(Figure 10 (c))”

Reviewer 2 Report

MECHANISM OF ANTIFUNGAL ACTION OF MONOTERPENE ISOESPINTANOL AGAINST CLINICAL ISOLATES OF Candida tropicalis

This research was to evaluate the antifungal activity of ISO, estimate its ability to eradicate mature biofilms and explore the mechanisms of action against clinical isolates of C. tropicalis, contributing to the search for new compounds of natural origin that can serve as adjuvants in the treatment of pathogenic yeasts resistant to antifungals. The work is interesting and the idea support in this search.

However, there are several points that should be resolved before publication.

The first thing I should note is that the sections and format are not organized as required by the journal author’s instructions, there seems to be no introduction, and the methods come before the results. Lines are not numbered for ease of review.

It would be interesting if introduction need to a little bit improved and increase. 

 Some minor comments

- Please check all names of the organisms and change to Italic

- Abstract and Conclusions: In order to entice readers, the Abstract and Conclusions should provide additional information.

All manuscript should be checked for typographical errors.

Thank you very much

Best Wishes
